# Establishing the Minimum Clinically Significant Difference (MCID) and the Patient Acceptable Symptom Score (PASS) for the Hospital Anxiety and Depression Scale (HADS) in Patients with Rotator Cuff Disease and Shoulder Prosthesis

**DOI:** 10.3390/jcm12041540

**Published:** 2023-02-15

**Authors:** Umile Giuseppe Longo, Rocco Papalia, Sergio De Salvatore, Andrea Marinozzi, Ilaria Piergentili, Alberto Lalli, Benedetta Bandini, Edoardo Franceschetti, Vincenzo Denaro

**Affiliations:** 1Research Unit of Orthopaedic and Trauma Surgery, Fondazione Policlinico Universitario Campus Bio-Medico, Via Alvaro del Portillo 200, 00128 Roma, Italy; 2Research Unit of Orthopaedic and Trauma Surgery, Department of Medicine and Surgery, Università Campus Bio-Medico di Roma, Via Alvaro del Portillo 21, 00128 Roma, Italy; 3Laboratory of Measurement and Biomedical Instrumentation, Campus Bio-Medico University, Via Alvaro del Portillo 200, 00128 Rome, Italy

**Keywords:** MCID, PASS, HADS, PROM, shoulder arthroplasty

## Abstract

Insufficiently treated shoulder pain may cause mental disturbances, including depression and anxiety. The Hospital Anxiety and Depression Scale (HADS) is a patient-reported outcome measure (PROM) that aims to identify depression and anxiety in patients in nonpsychiatric wards. The aim of this study was to identify the minimum clinically important difference (MCID) and patient acceptable symptom state (PASS) scores for the HADS in a cohort of individuals with rotator cuff disease. Using the HADS, participants’ degrees of anxiety and depression were assessed at inception and at their final assessment 6 months after surgery. To calculate the MCID and the PASS, distribution and anchor approaches were employed. The MCID from inception to final assessment was 5.7 on the HADS, 3.8 on the HADS-A, and 3.3 on the HADS-D. A 5.7 amelioration on the HADS score, 3.8 on the HADS-A, and 3.3 on the HADS-D, from inception to final assessment, meant that patients had reached a clinically meaningful improvement in their symptom state. The PASS was 7 on the HADS, 3.5 on the HADS-A, and 3.5 on the HADS-D; therefore, for the majority of patients, a score of at least 7 on the HADS, 3.5 on the HADS-A, and 3.5 on the HADS-D at final evaluation was considered a satisfactory symptom state.

## 1. Introduction

Of patients complaining of shoulder pain during outpatient visits, 70% are diagnosed with a rotator cuff tear (RCT) [1].

RCTs can be successfully surgically treated via shoulder arthroplasty [2]. The number of shoulder arthroplasty patients has risen over the past few decades [3,4,5]. Shoulder arthroplasty, hemiarthroplasty, and reverse total shoulder arthroplasty represent the most common prosthetic options [6].

It has been demonstrated that shoulder diseases are linked to night discomfort, insomnia, and sleeplessness as a result of pain in the afflicted arm [7,8], and in severe cases might even be detrimental to day-to-day activity [9], which leads the patient to consider surgery [10].

Approximately 25% of patients with RCTs experience anxiety or depression, and psychological health may be a key indicator of how well a patient will recover from arthroscopic rotator cuff surgery [11,12,13]. However, in clinical practice functional evaluation is often concentrated on the objective elements of the disease, such as assessing range of motion and strength [14,15].

For this reason, orthopaedic research has experienced a transformation, and the development of established patient-oriented metrics has given clinical outcome evaluation a fresh viewpoint, in addition to objective measurements [16].

Patient-reported outcome measures (PROMs) are widely employed to evaluate patients’ wellbeing [17]. PROMs are subjective patient-reported scales developed to provide health status outcomes, bypassing a clinician’s interpretation [18,19].

The Hospital Anxiety and Depression Scale (HADS) is an example of a PROM that aims to identify depression and anxiety in patients in nonpsychiatric wards [20].

This questionnaire does not include physical indications of emotional discomfort, such as headaches, weight loss, and sleeplessness, which can result from an underlying medical condition rather than from emotional distress [21]. The HADS has been widely utilized as a screening tool to assess the psychological conditions of patients with musculoskeletal illnesses [20,22,23], as patients with shoulder discomfort who receive insufficient pain relief may develop mental disturbances, including depression and anxiety [24].

In fact, to assess the efficacy of a clinical treatment, the difference between PROMs before and after an intervention is computed [25]. This variance may indicate an improvement in quality of sleep after a RCT repair, but may not quantify the benefit [26]. A genuine difference for a patient might not be correlated to a clinically relevant difference [27]. Therefore, it is necessary to find a real value at which a patient could experience a real beneficial change [28].

The minimum clinically important difference (MCID) was defined to solve this problem [27]. Designating a difference threshold for clinical relevance, the MCID seeks to close the gap between numerical data and patient experience [29,30].

The patient acceptable symptom state (PASS) constitutes the lowest PROM threshold that correlates to an acceptable wellness of a patient [31]. Patients consider themselves healthy beyond this value [31,32].

The MCID is the lowest variation in health condition detected as significant by patients. The PASS establishes the symptom threshold at which individuals believe their health to be at a satisfactory level. Therefore, the MCID correlates to “feeling better” while the PASS correlates to “feeling well” [33,34].

The aim of this study was to find the minimum clinically important difference (MCID) and patient acceptable symptom state (PASS) scores for the HADS in a cohort of individuals with rotator cuff disease. According to the authors’ understanding, this is the first research to identify the MCID and PASS values for the HADS, HADS-A and HADS-D of patients with rotator cuff disease.

## 2. Materials and Methods

### 2.1. Participants

This study retrieved data from 55 patients (28 men and 27 women, mean age 61.5 ± 11.2 years) with primary RCT, prospectively enrolled in a six-month program at the Campus Bio-Medico of Rome between January 2019 and December 2019. Baseline and follow-up paper-based HADS scores, Oxford shoulder score (OSS) data and short-form health survey (SF-36) responses, before and after rotator cuff repair, were retrieved.

### 2.2. Hospital Anxiety and Depression Scale (HADS)

Using the HADS, participants’ degrees of anxiety and depression were assessed before and 6 months after intervention. The 14-item HADS is a self-assessment tool that measures patients’ subjective levels of anxiety and depression [35]. It consists of two 7-item anxiety (HADS-A) and depression (HADS-D) subscales, each summed to yield values from 0 to 21; a higher score depicts a worse condition. The total HADS value ranges from 0 (best condition) to 42 (worst condition).

### 2.3. Statistical Analysis

With a 0.4 Cohen’s d effect size of HADS-A, as per the literature [36], alpha = 0.05 and power = 0.8, a cohort of 41 patients was considered essential. Data normality for the HADS, HADS-A, and HADS-D were calculated using the Shapiro–Wilk test of normality. As the normal distributions, data were left unchanged. Inception and final follow-up values were compared using paired-sample T-tests. The statistical level of significance was 0.05. All data were analysed using SPSS (version 26; IBM Corp., Armonk, NY, USA).

### 2.4. Calculation of the MCID

To calculate the MCID for the HADS, HADS-A, and HADS-D, both distribution and anchor approaches were used. Several distribution-based techniques that were employed: 0.5 standard deviation (0.5 SD), the standard error of measurement (SEM), and the minimum detectable change (MDC).

A 0.5 SD was correlated to effect size (0.5 SD was a median effect), the MDC reflected the lowest variation beyond the measurement error with a confidence interval (usually 95% confidence), whereas the SEM was the lowest variation beyond the measurement error. The Cronbach’s alpha was applied as an estimate of reliability of the HADS, HADS-A, and HADS-D to compute the SEM and MDC.

To calculate the MCID using the anchor approach, patients answered the following question about improvement at six-month follow-ups “How do you feel following the surgical procedure?” The potential responses were “much worse”, “slightly worse”, “equal”, “slightly better”, and “much better”.

Participants who answered “much worse”, “slightly worse”, and “equal” were considered no responders. Participants who answered “slightly better” were considered minimally improved.

To assess the consistency of results, another anchor approach question was used: “How would you rate your general health today compared to a year ago?” The possible responses were “much better”, “somewhat better”, “about the same”, “somewhat worse”, and “much worse”.

Participants who answered “about the same”, “somewhat worse”, and “much worse” were considered no responders.

Participants who answered “somewhat better” were considered minimally improved.

Two anchor approaches were applied. The receiver operating characteristics (ROC) curve approach compared no responders with the minimally improved. The Youden index was taken into consideration as the best cut-off value for each dimension. An area under the curve (AUC) of 1 was considered perfect, whereas an AUC of 0.5 was considered no better than chance, 0.7 was fair, 0.8 was good, and 0.9 was excellent [37]. Given this, values of AUC above 0.5 were considered valid and values of AUC higher than 0.7 were considered acceptable. Another anchor approach was considering if a patient’s mean change score was minimally improved.

### 2.5. Calculation of PASS

Kvien et al. [34] suggested the following question as an anchor for a PASS “Do you believe your current condition to be satisfactory in light of all the activities you engage in on a daily basis, your degree of discomfort, and your functional impairment?” As no such question was available for this study, to calculate the PASS for the HADS, HADS-A and HADS-D, we asked, “In general, would you say your health is at least good?” The possible answers were “yes” or “no”; patients who responded “yes” were classified as having a sufficient symptom state. To assess the consistency of results, another question was asked: “Has shoulder pain prevented you from doing your regular work (including housework)?” Patients could answer “yes” or “no”; individuals who responded “no” were considered to have a tolerable symptom status. These questions are both valid; in fact, according to the study carried out by Kvien and colleagues, [34] a valid anchor for a PASS is one that considers pain, physical function, and patients’ contentment.

The 75th percentile score of individuals who believe their symptoms to be tolerable is usually used to define a PASS [34]. Another common method of determining a PASS cut-off is to locate the position on the ROC curve where the threshold has been found by applying the Youden index.

## 3. Results

HADS, HADS-A, and HADS-D scores were defined as normally distributed using the Shapiro–Wilk test of normality (*p* > 0.05). At inception, HADS index values were between 0 and 32 (0% floor and 6.9% ceiling, respectively) with a mean value of 10.6 ± 8.1. At six-month follow-ups, HADS index values were between 0 and 27 (0% floor and 32.7% ceiling, respectively) with a mean value of 4.4 ± 6.3. A statistically meaningful difference from inception to final follow-up was identified (*p* < 0.001).

At inception, HADS-A index values were between 0 and 16 (0% floor and 13% ceiling, respectively) with a mean value of 5.9 ± 4.4. At six-month follow-ups, HADS-A index values were between 0 and 13 (0% floor and 36.2% ceiling, respectively) with a mean value of 2.7 ± 3.2. A statistically meaningful difference from inception to final follow-up was identified (*p* < 0.001).

At inception, HADS-D index values were between 0 and 18 (0% floor and 29.3% ceiling, respectively) with a mean value of 4.7 ± 4.1. At six-month follow-ups, HADS-A index values were between 0 and 16 (0% floor and 62.1% ceiling, respectively) with a mean value of 1.7 ± 3.3. A statistically meaningful difference from inception to final follow-up was identified (*p* < 0.001).

The anxiety (α = 0.8) and depression (α = 0.8) subscales of this study’s internal consistency reliability were satisfactory. The internal consistency reliability of the global HADS results were sufficient as well (α = 0.9).

### 3.1. Thresholds of MCID

As the AUC of the ROC analysis using “How do you feel following the surgical procedure?” as the anchor was between 0.4 and 0.6, this anchor was not considered valid.

MCID measurements for HADS index values were between 1.8 and 5.7 (Table 1). We defined an MCID of 2.9 (0.5 SD) with a medium effect size (ES = 0.5), an MCID of 1.8 (SEM) with an internal consistency reliability of 0.9, and an MCID of 4.9 (MDC) with a 95% confidence level. The ROC method found an MCID of 2.5 (AUC = 0.7) and the MC method found an MCID of 5.7.

MCID measurements for HADS-A index values were between 1.4 and 3.8. We defined an MCID of 1.7 (0.5 SD) with a medium effect size (ES = 0.5), an MCID of 1.4 (SEM) with an internal consistency reliability of 0.8, and an MCID of 3.8 (MDC) with a 95% confidence level. The ROC method found an MCID of 3.5 (AUC = 0.6) and the MC method found an MCID of 2.8.

MCID measurements for HADS-D index values were between 1.2 and 3.3. We defined an MCID of 1.5 (0.5 SD) with a medium effect size (ES = 0.5), an MCID of 1.2 (SEM) with an internal consistency reliability of 0.8, and an MCID of 3.3 (MDC) with a 95% confidence level. The ROC method found an MCID of 1.5 (AUC = 0.8) and the MC method found an MCID of 2.9.

### 3.2. Thresholds of PASS

PASS measurements calculated for the HADS ranged from 3.5 to 7. The HADS threshold that provided the highest levels of specificity and sensitivity to detect a PASS was 7 (AUC = 0.9 and AUC = 0.8). The cut-offs calculated using the 75th percentile approach were 3.5 and 4 (Table 2).

PASS measurements calculated for the HADS-A ranged from 3 to 4.5. The HADS thresholds that provided the highest levels of specificity and sensitivity to detect a PASS were 3.5 (AUC = 0.9) and 4.5 (AUC = 0.8). The cut-offs calculated using the 75th percentile approach were 3 and 3.8 (Table 2).

PASS measurements calculated for the HADS-D ranged from 1 to 3.5. The HADS thresholds that provided the highest levels of specificity and sensitivity to detect a PASS were 3.5 (AUC = 0.9) and 2.5 (AUC = 0.8). The cut-offs calculated using the 75th percentile approach was 1 with both anchors (Table 2).

## 4. Discussion

The primary finding of this study is that the MCID from inception to final follow-up after 6 months was 5.7 for the HADS, 3.8 for the HADS-A, and 3.3 for the HADS-D, whereas the PASS was 7 for the HADS, 3.5 for the HADS-A and 3.5 for the HADS-D. These scores represent a clinically meaningful improvement in patients’ symptom states.

According to the authors’ understanding, this is the first study to identify MCID and PASS values for the HADS, HADS-A and HADS-D of patients with rotator cuff disease.

To the authors’ knowledge, four articles have found MCID values for the HADS, HADS-A and HADS-D. However, their cohort comprised only patients suffering from pulmonary or cardiovascular disease. Puhan et al. [38] reported MCIDs of 1.41 and 1.57 for the HADS anxiety score and 1.68 and 1.60 for the HADS total score of patients with chronic obstructive pulmonary disease (COPD). Smid et al. [39] calculated MCIDs between 1.1 and 2 for the HADS-A and between 1.4 and 1.8 for the HADS-D in COPD patients. Lemay et al. [40] showed MCIDs between 0.81 and 5.21 for the HADS-A and between 0.5 to 5.57 for the HADS-D, and an MCID of 1.7 points for the HADS of patients with cardiovascular diseases. Wynne et al. [41] reported MCIDs of 2 points for both the HADS-A and HADS-D in patients with bronchiectasis.

In this study, 15 MCIDs were found: between 1.8 and 5.7 for the total HADS, between 1.4 and 3.8 for the HADS-A, and between 1.2 and 3.3 for the HADS-D.

As stated by Copay et al. [42], a genuine MCID should be at least higher than the SEM value, and also reflect the patient’s view of the significance of the change. Moreover, as found by Stipancic et al. [43], to be acceptable, an MCID should be greater than the associated MDC. These factors indicate that the MC approach is more suitable for the total HADS; as the thresholds calculated using ROC and MC methods for the HADS-A and HADS-D were less than their respective MDCs, these methods were not appropriate. Therefore, the MCID of the HADS was 5.7, the MCID of the HADS-A was 3.8, and the MCID of the HADS-D was 3.3.

Applying the 75th percentile and ROC approaches using two anchors, 12 PASS thresholds were found. The cut-offs for the “In general, would you say your health is at least good?” anchor were 7 and 3.5 for the HADS, 3.5 and 3 for the HADS-A, and 3.5 and 1 for the HADS-D, using the ROC and 75th percentile approaches, respectively. The cut-offs for the “Has shoulder pain prevented you from doing your regular work (including housework)?” anchor were calculated as 7 and 4 for the HADS, 4.5 and 3.8 for the HADS-A, and 2.5 and 1 for the HADS-D, using the ROC and 75th percentile approaches, respectively.

The AUC calculated using the “In general, would you say your health is at least good?” anchor (AUC = 0.9) was always higher than the AUC calculated using the “Has shoulder pain prevented you from doing your regular work (including housework)?” anchor (AUC = 0.8). For this reason, the first question seemed to be most appropriate. The ROC method was most commonly adopted in the literature [44,45,46,47,48]. Given these considerations, the most appropriate PASS values seemed to be 7 for the HADS, 3.5 for the HADS-A, and 3.5 for the HADS-D.

This study had several strengths. According to the authors, no other study in the literature provides both the MCID and PASS of the HADS, HADS-A, and HADS-D for patients with rotator cuff disease. In addition, the questions used as anchors were reliable and frequently used in the literature; in fact, the MCID and PASS were computed using the most effective ad hoc techniques. Moreover, as two anchors were accessible for MCID and PASS values calculation, data coherence using many anchors was evaluated, providing this study a greater reliability of results. Finally, the analysis was conducted not only for the global HADS, but also for its dimensions, anxiety and depression, which further enhanced the applicability of this study’s results.

This study also had some limitations. The scores were computed at a final follow-up of 6 months, and therefore did not provide information for longer-term conditions and results. It is possible that the MCID scores varied after the final evaluation. This highlights the need for further studies involving longer follow-up, in order to deliver acceptable long-term data on the topic. Furthermore, even if the retrieved cohort was sufficient, as supported by the a priori power analysis, more patients have frequently been studied in the literature. The authors suggest that further studies involve larger cohorts.

## 5. Conclusions

This study found that patients showed a clinically significant improvement in their symptoms, as measured using the HADS, HADS-A, and HADS-D scores. The MCID, or minimum clinically important difference, was 5.7 for the HADS, 3.8 for the HADS-A, and 3.3 for the HADS-D. This means that patients had an average improvement of 5.7 on their HADS scores, 3.8 on their HADS-A scores, and 3.3 on their HADS-D scores from the beginning to the end of the assessment period.

Additionally, this study found that the majority of patients achieved a satisfactory symptom state, as determined using the PASS score. A HADS score of 7, a HADS-A score of 3.5, and a HADS-D score of 3.5 at the final evaluation were considered a positive outcome. These results suggest that patients made substantial progress in managing their symptoms over the course of the study.

## Figures and Tables

**Table 1 jcm-12-01540-t001:** Minimum clinically important difference (MCIDs) of the Hospital Anxiety and Depression Scale (HADS), anxiety (HADS-A), and depression (HADS-D).

Anchor				Improvement Question 1 ^1^		Improvement Question 2 ^2^	
**Score**	0.5 SD	SEM	MDC	ROC (AUC)	MC	ROC (AUC)	MC
**HADS**	2.9	1.8	4.9	10.5 (0.5)	5.8	2.5 (0.7)	5.7
**HADS-A**	1.7	1.4	3.8	7 (0.4)	2.7	3.5 (0.6)	2.8
**HADS-D**	1.5	1.2	3.3	4.5 (0.6)	3.1	1.5 (0.8)	2.9

^1^ “How do you feel following the surgical procedure?” ^2^ “How would you rate your general health today compared to a year ago?”

**Table 2 jcm-12-01540-t002:** PASS measurements for the HADS, HADS-A, and HADS-D.

Score	ROC (AUC)	75th PERCENTILE	ANCHOR
**HADS**	7 (0.9)	3.5	“In general, would you say your health is at least good?”
	7 (0.8)	4	“Has shoulder pain prevented you from doing your regular work (including housework)?”
**HADS-A**	3.5 (0.9)	3	“In general, would you say your health is at least good?”
	4.5 (0.8)	3.8	“Has shoulder pain prevented you from doing your regular work (including housework)?”
**HADS-D**	3.5 (0.9)	1	“In general, would you say your health is at least good?”
	2.5 (0.8)	1	“Has shoulder pain prevented you from doing your regular work (including housework)?”

## Data Availability

The data presented in this study are available on request from the corresponding author.

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
