# Peer review of "Establishing the Minimum Clinically Significant Difference (MCID) and the Patient Acceptable Symptom Score (PASS) for the Hospital Anxiety and Depression Scale (HADS) in Patients with Rotator Cuff Disease and Shoulder Prosthesis"

_jcm, 2023, doi:10.3390/jcm12041540_

Round 1

Reviewer 1 Report

We should know if patients have reached a clinically meaningful improvement in their symptom state - after our treatment.

So, this work is important.

But the discussion and conclusions are written unclearly, it is very difficult to follow the authors. 

Author Response

Dear reviewer,

We would like to thank you for the helpful comments and suggestions. We have revised the paper accordingly and hope that the work is now ready for publication. The changes are itemized below with our comments to the reviewer’s suggestions. Changes made in the text are highlighted in yellow in the original manuscript.

Reviewer #1:

We should know if patients have reached a clinically meaningful improvement in their symptom state - after our treatment.

  • Thank You for the comment, we modified the manuscript accordingly: Lines 213-218: “The main finding of this study is that the MCID from inception to final follow-up after 6 months is 5.7 for HADS, 3.8 for HADS-A and 3.3 for HADS-D, while the PASS is 7 for HADS, 3.5 for HADS-A and 3.5 for HADS-D. These scores represent a clinically meaningful improvement in the patients’ symptom state.”

So, this work is important. But the discussion and conclusions are written unclearly, it is very difficult to follow the authors.

  • Thanks for the comment. We revised the discussion and the conclusion accordingly.

Reviewer 2 Report

Line 22: finedidentify

Line 25-26: delete:  All data were 26 analysed by SPSS (version 26; IBM Corp)

Line71: applied defined

Line 81: finding  defining

Line 81-85: states two times the same, merge the two lines and mention that the MICD for HADS has not been defined yet which is you primairy aim/goal.

Line 88: article  study

Line 88-92: please describe the study population in more detail. Baseline characteristics (population, pathology, etc.). Which intervention was conducted and were there any complications? Retears in case of cuff repair? Not clear to me which intervention..  

How was the questionnaire collected (paper based? digital? Etc..)

How was the enorollment?

Retrospective? Prospective?

Line 95: why did you use 6 months? Please clarify.

Line 106: by using

Methods section needs to be clarified in order to make this study ready for publication. This will need a revision from my point of view. 

Because of this I would say it needs a major revision.  

Author Response

Dear reviewer,

We would like to thank you for the helpful comments and suggestions. We have revised the paper accordingly and hope that the work is now ready for publication. The changes are itemized below with our comments to the reviewer’s suggestions. Changes made in the text are highlighted in yellow in the original manuscript.

Reviewer #2:

Line 22: fined  identify

Thank You for the comment, we modified the manuscript accordingly.

Line 25-26: delete: All data were 26 analysed by SPSS (version 26; IBM Corp)

Thank You for the comment, we modified the manuscript accordingly.

Line71: applied  defined

Thank You for the comment, we modified the manuscript accordingly.

Line 81: finding  defining

Thank You for the comment, we modified the manuscript accordingly.

Line 81-85: states two times the same, merge the two lines and mention that the MICD for HADS

has not been defined yet which is you primary aim/goal.

Thank You for the comment, we modified the manuscript accordingly.

Line 88: article  study

Thank You for the comment, we modified the manuscript accordingly.

Line 88-92: please describe the study population in more detail. Baseline characteristics

(population, pathology, etc.). Which intervention was conducted and were there any

complications? Retears in case of cuff repair? Not clear to me which intervention.

How was the questionnaire collected (paper based? digital? Etc..)

How was the enrollment? Retrospective? Prospective?

Thank You for the comment, we modified the manuscript accordingly:

Lines 87-91: “This study retrieved data from 55 patients (28 men and 27 women, mean age 61.5 ± 11.2 years) with primary RCT, prospectively enrolled in a 6-months program at the Campus Bio-Medico of Rome between January 2019 and December 2019, retrieving baseline and follow-up paper-based HADS, Oxford Shoulder Score (OSS) data and Short Form Health Survey (SF-36) before and after rotator cuff repair”.

Line 95: why did you use 6 months? Please clarify.

  • Thank You for the comment. We used 6 months of follow up to represent the most valuable change in this score as reported in other studies (PMID: 33915704, 33668868, 34444415. We modified the manuscript accordingly:

Lines 265-269: “The scores were computed at a final follow-up of 6 months, and therefore does not give information for the longer-term conditions and results. It is possible that the MCID scores may vary after the set final evaluation. This highlights the need for further studies involving longer follow-up, in order to deliver acceptable long-term data on the topic”.

Line 106: by using

Thank You for the comment, we modified the manuscript accordingly.

Round 2

Reviewer 2 Report

The authors have made the requited changes. It will only need a minor spell check.